# Physiological, Biochemical, and Molecular Mechanisms of Resistance of *Poacynum hendersonii* to *Melampsora apocyni*

**DOI:** 10.3390/plants14162589

**Published:** 2025-08-20

**Authors:** Junjun Gu, Endong Shang, Miao Ma

**Affiliations:** Key Laboratory of Oasis Town and Mountain-basin System Ecology, Key Laboratory of Xinjiang Phytomedicine Resource Utilization, Ministry of Education, College of life Sciences, Shihezi University, Shihezi 832003, China; 20232306208@stu.shzu.edu.cn (J.G.); 20232106042@stu.shzu.edu.cn (E.S.)

**Keywords:** rust, transcriptomics, phytohormones, phenylpropanoid metabolism, lignin, antioxidant activity

## Abstract

The rust disease caused by *Melampsora apocyni* seriously affects the growth of *Poacynum hendersonii.* However, the defense mechanisms against rust infection remain unclear. This study explored the regulatory mechanisms of *P. hendersonii* in response to rust disease through combined physiological, biochemical, and transcriptomic analyses. The results showed that with the increase in disease severity, the chlorophyll content of leaves decreased significantly, while the antioxidant and phenylalanine ammonia lyase activities progressively increased. Mild infection triggered an 11.9-fold surge in salicylic acid levels and a sharp decline in abscisic acid compared to controls, as well as increased synthesis of total phenolics, total flavonoids, chlorogenic acid, cryptochlorogenic acid, isoquercetin, hyperoside, rutin, and astragalin. Transcriptome analysis showed that the “plant–pathogen interaction, plant hormone signal transduction and phenylpropanoid biosynthesis” pathways were significantly up-regulated in the mild infection stage, while “glycerophospholipid metabolism, fatty acid degradation and ABC transporters” were activated in the severe infection stage. In summary, *P. hendersonii* regulates energy metabolism and phenylpropanoid metabolism through salicylic acid signaling and promotes the accumulation of secondary metabolites and the lignification process of leaves, thereby enhancing rust resistance. Key enzyme genes (COMT, POD, CAD, F5H) and metabolites (chlorogenic acid, isoquercitrin, rutin) can be used as important targets for disease resistance breeding. Our research provides important reference for the prevention and control of *M. apocyni* in *P. hendersonii*.

## 1. Introduction

*Poacynum hendersonii*, an erect semi-shrub belonging to the Apocynaceae family [1,2], is primarily distributed across China, Central Asia, and Russia, and it is a dominant species in saline lowland meadows [3,4]. Its leaves are rich in flavonoids [5,6], which have been shown to lower blood lipids [7], reduce blood pressure [8,9], decrease blood sugar levels [10], and possess anti-anxiety effects [11], making it a popular choice for traditional health teas [12,13,14]. Recent studies have shown that the flavonoids in this plant also exhibit biological activities, including the inhibition of tumor cell proliferation [15], antioxidant properties [16,17], and antithrombotic activity. Consequently, it has found widespread application in the fields of medicine. Furthermore, the bast fibers from its stems possess high gloss and strength [18,19], making them a premium natural textile material favored by many. Additionally, the significant accumulation of lithium in its leaves endows the plant with important industrial economic value [20,21]. However, the large-scale conversion of saline meadows into farmland and the overharvesting of stem and leaf resources have led to a rapid decline in the number and scale of wild populations [22]. To alleviate this threat, the artificial cultivation of *P. hendersonii* to natural populations has been initiated in various regions in recent years [23].

Although the productivity of *P. hendersonii* has significantly increased under artificial cultivation in recent years, high-density planting has also led to a marked rise in the frequency of crop diseases, which is much higher than that observed in natural populations. Among these diseases, leaf rust caused by *Melampsora apocyni* is particularly severe, with individual infection rates reaching up to 99% [24]. The symptoms of this disease manifest as numerous golden-yellow uredinial pustules on the leaf surface, leading to yellowing and shedding of the leaves. In severe cases, the entire plant may wilt and die [25]. This disease has become a major bottleneck restricting the sustainable development of *P. hendersonii* in the pharmaceutical, tea, and fiber textile industries.

Currently, methods such as the removal of diseased plants and the regulation of water and fertilizer can effectively reduce the incidence of rust disease. However, these approaches are associated with high operational costs and limited efficacy in large-scale cultivation. While chemical control can temporarily alleviate the symptoms of rust disease, prolonged application may lead to increased pathogen resistance and the risk of pesticide residues exceeding safety limits, which severely restricts the development and utilization of *P. hendersonii* as a dual-purpose crop for food and medicine. When plants are attacked by pathogenic microorganisms, they often initiate a series of physiological and biochemical reactions to enhance their resistance against pathogens. Previous studies have only focused on the identification of pathogen pathogenicity and the statistical analysis of disease incidence in the field [24], without delving into the host plant’s own disease resistance mechanisms.

To elucidate the defense mechanism of *P. hendersonii* against rust disease, this study investigates the effects of rust disease on its photosynthetic pigment content, antioxidant and defense enzyme activities, endogenous hormone levels, secondary metabolite content, and gene expression, and it identifies key disease-resistant genes and metabolites, providing a theoretical basis for the cultivation of rust-resistant germplasm resources.

## 2. Results

### 2.1. The Effect of Rust Disease on the Antioxidant Activity in Leaves of P. hendersonii

Rust disease significantly impacts the activities of superoxide dismutase (SOD) and polyphenol oxidase (PPO), as well as the total antioxidant capacity (FRAP) in P. hendersonii (Figure 1). Notably, SOD activity in severely infected leaves increased by 9% compared to mildly infected leaves (Figure 1A). The PPO activity displayed a trend of initially increasing and then decreasing with the severity of infection, showing increases of 134.4% and 59.1% under mild and severe infection conditions, respectively, compared to the control group (CK) (Figure 1B). Similarly, FRAP exhibited an upward trend, with increases of 77.4% and 214.5% under mild and severe infection conditions, respectively, in comparison to the CK group (Figure 1C). Furthermore, rust disease infection did not significantly affect the H_2_O_2_ content (Figure 1D).

### 2.2. The Effect of Rust Disease on Chlorophyll Content in Leaves of P. hendersonii

Infection with rust significantly reduced the chlorophyll content in the leaves of *P. hendersonii* (Figure 2). Compared to the CK group, the contents of chlorophyll a and chlorophyll b decreased by 47.43% and 24.56%, respectively, under mild infection conditions (Figure 2A); Under severe infection conditions, the contents of chlorophyll a and chlorophyll b decreased by 75.44% and 63.01%, respectively, compared to the CK group (Figure 2B).

### 2.3. The Effect of Rust Disease on Hormone Content in Leaves of P. hendersonii

Rust significantly influences the concentrations of salicylic acid (SA) and abscisic acid (ABA) in the leaves of *P. hendersonii*. Compared to the CK group, mild infection significantly increased the concentration of SA in *P. hendersonii*, reaching 11.9 times that of the control group (Figure 3A), while reducing the content of ABA to 41.2% of that in the CK group (Figure 3B). Under conditions of severe infection, the SA concentrations significantly decreased compared to mild infection but remained significantly higher than that of the CK group. Conversely, the ABA content showed a significant increase compared to mild infection (M), although no significant difference was observed when compared to the CK group.

### 2.4. Effects of Rust Disease on Secondary Metabolites in the Leaves of P. hendersonii

Rust disease significantly impacted the activity of phenylalanine ammonia-lyase (PAL) (Figure 4A), as well as the levels of total phenols (Figure 4B) and total flavonoids (Figure 4C) in the leaves of *P. hendersonii*. Compared to the CK group, PAL activity in rust-infected leaves showed a substantial increase, with PAL activity rising by 74.39% in mildly infected leaves and by 138.67% in severely infected leaves relative to the CK group. The content of total phenols and total flavonoids displayed a trend of initial increase followed by a decrease: under mild and severe infection levels, total phenol content reached 123.47% and 96.3% of the control group, respectively, while total flavonoid content was 167.67% and 78.18% of the CK group, respectively.

As the severity of the disease in *P. hendersonii* increases, the contents of chlorogenic acid, cryptochlorogenic acid, isoquercetin, hypericin, rutin, and astragaloside in the leaves initially rise before subsequently declining. In contrast, the levels of quercetin and kaempferol consistently decrease (Figure 5). Under mild disease conditions, the concentrations of chlorogenic acid, cryptochlorogenic acid, isoquercetin, hypericin, rutin, and astragaloside increased by 90.7%, 78.6%, 180.5%, 53.4%, 1154.6%, and 182.4%, respectively, compared to the control group.

### 2.5. Transcriptome Analysis

We conducted a transcriptomic analysis of leaf samples from *P. hendersonii* infected with rust disease, resulting in the construction of nine cDNA libraries (Table 1). After sequencing, we obtained 184,965,780 clean reads, which accounted for 97% of the total raw reads. The percentage of Q20 bases was above 97.94%, while the percentage of Q30 bases exceeded 94.04%. The GC content of the bases ranged from 44.32% to 45.15%.

The correlation index R^2^ among samples within the treatment group is consistently above 0.875 (Figure 6A), indicating that the reliability of biological replicates meets the requirements for subsequent analyses. Principal Component Analysis (PCA) reveals that PC1 and PC2 together account for 91.26% of the variance, with a significant separation observed between the control group (CK) and the rust disease-infected group (M/S) (Figure 6B). This confirms that pathogen infection leads to a reprogramming of the transcriptome in *Macadamia integrifolia*. Hierarchical clustering analysis further demonstrates that samples with mild (M) and severe (S) disease cluster together (Figure 6C), suggesting a convergence in gene expression patterns during disease progression. Compared to the CK group, a total of 5902 differentially expressed genes (DEGs) were identified in the M group (2557 upregulated and 3345 downregulated). In the S group, a total of 4752 DEGs were identified compared to the CK group (2006 upregulated and 2746 downregulated). When compared to the M group, a total of 3537 DEGs were identified in the S group (1849 upregulated and 1688 downregulated) (Figure 6D).

To further elucidate the biological functions of differentially expressed genes (DEGs), we conducted Gene Ontology (GO) functional classification on the identified DEGs. In comparison to the control group (CK), the M group exhibited 15 annotations in the biological process category and 18 in the molecular function category. The top 10 GO terms in the biological process category included the carbohydrate metabolic process (161), polysaccharide metabolic process (40), glucan metabolic process (33), cellulose metabolic process (17), cellulose biosynthetic process (17), and transmembrane transport process (193), with the highest number of genes related to transmembrane transport. The top 10 GO terms in the molecular function category included monooxygenase activity (111), oxidoreductase activity (111), acting on paired donors (110), incorporation or reduction of molecular oxygen (80), iron ion binding (121), UDP glycosyltransferase activity (121), heme binding (183), porphyrin binding (100), transmembrane transporter activity (184), and hydrolase activity (103) (Figure 7A). Compared to the CK group, the top 10 GO terms in the biological process and molecular function categories for DEGs in the S group were largely consistent with those in the M group (Figure 7B).

We conducted a KEGG pathway analysis of differentially expressed genes (DEGs). Compared to the control group (CK), a total of 1134 DEGs in the treatment group (M) were annotated across 124 metabolic pathways. Among these, several pathways related to the biosynthesis of various secondary metabolites in plants, starch and sucrose metabolism, biosynthesis of cutin, suberin, and wax, maize protein biosynthesis, and plant–pathogen interactions showed significant enrichment (Figure 8A). By comparing the gene enrichment across pathways, we observed that most DEGs in the flavonoid biosynthesis and carbon fixation pathways of photosynthetic organisms were significantly downregulated. In contrast, the majority of DEGs in the pathways related to starch and sucrose metabolism, biosynthesis of cutin, suberin, and wax, plant–pathogen interactions, α-linolenic acid metabolism, plant hormone signal transduction, and phenylpropanoid biosynthesis were significantly upregulated. These upregulated metabolic pathways may play a crucial role in the defense response of *P. hendersonii* during mild infection stages (Table 2).

Compared to the control group (CK), a total of 932 differentially expressed genes (DEGs) in the S group were annotated to 124 metabolic pathways. Among these, the enrichment of pathways related to starch and sucrose metabolism, biosynthesis of various plant secondary metabolites, and plant hormone signal transduction was most pronounced (Figure 8B). By comparing the number of enriched DEGs across each pathway, we observed that most DEGs in pathways such as carbon metabolism, carbon fixation in photosynthetic organisms, and flavonoid biosynthesis were significantly downregulated. In contrast, most genes in pathways such as phenylpropanoid biosynthesis, glycerophospholipid metabolism, fatty acid degradation, and ABC transport showed upregulation. These upregulated metabolic pathways may play a crucial role in the defense response of *P. hendersonii* during the severe disease phase (Table 3).

The biosynthetic pathway of phenylpropanoids involves the synthesis of lignin, phenolic compounds, and flavonoids, which are crucial components of plant defense mechanisms. Under mild disease conditions, the differentially expressed genes (DEGs) in this pathway are mainly related to the synthesis of the following enzymes: caffeic acid 3-O-methyltransferase (COMT, EC 2.1.1.68), peroxidase (POD, EC 1.11.17), and cinnamate-4-hydroxylase (CAD, EC 1.1.1.195). Among them, the DEG g354, which is involved in COMT synthesis, shows a significantly upregulated expression, with a log_2_(FC) value of 5.4; the DEGs g17904, g11867, novel.932, g13084, g14106, g19451, g944, g15458, novel.465, g806, and g17504 involved in POD synthesis also exhibit upregulated expression, with log_2_(FC) values of 3.6, 10.2, 5.7, 3.7, 1.5, 1.7, 3.3, 3.7, 4.6, 7.7, and 1.6, respectively; and the DEGs g4187, g16782, and g15449 involved in CAD synthesis also show upregulated expression, with log_2_(FC) values reaching 3.7, 1.9, and 1.8, respectively (Figure 9).

Under conditions of severe stress, the DEGs in the biosynthetic pathway of phenylpropanoids are primarily associated with peroxidase (POD, EC 1.11.17), cinnamate-4-hydroxylase (CAD, EC 1.1.1.195), and ferulate-5-hydroxylase (F5H). Among these, the DEGs involved in POD synthesis, namely g19451, g17904, g13084, g11867, g14106, novel.932, g806, g15458, g7667, g7131, g14878, and g2702, exhibit upregulation, with log_2_(FC) values of 2.2, 5.4, 4.9, 10.9, 2.2, 7.1, 11.5, 3.1, 2.8, 2.4, 5.1, and 1.9, respectively. The DEGs involved in CAD synthesis, g4187, g4197, g16935, and g16785, also show upregulation, with log_2_(FC) values of 3.4, 2.4, 3.2, and 4.3, respectively. Furthermore, the DEGs involved in F5H synthesis, g4395 and g4401, also exhibit upregulation, with log_2_(FC) values of 3.8 and 5.1, respectively (Figure 10).

To validate the RNA-seq data, we randomly selected five genes (g13063, g15449, g17904, g19451, g4187) for qRT-PCR analysis to determine their expression patterns. The results indicated that although there were differences in the gene expression levels detected by qRT-PCR and RNA-seq, the expression trends were generally consistent between the two methods, confirming the reliability of the RNA-seq results (Figure 11).

## 3. Discussion

Photosynthesis is a crucial physiological process for plant growth and biomass accumulation [26]. This study found that rust disease infection significantly reduced the contents of chlorophyll a and b in the leaves of *P. hendersonii*, with the chlorophyll content in severely infected leaves dropping to 18.03% (chlorophyll a) and 36.99% (chlorophyll b) of that in the control group, respectively. Combined with the results from the KEGG analysis, which show a continuous suppression of the ‘carbon fixation in photosynthetic organisms’ pathway, these findings confirm that the photosynthetic capacity of infected plants significantly declines. Notably, while photosynthesis is inhibited, significant changes in metabolic pathways occur: under mild infection conditions, the starch/sucrose metabolic pathway is upregulated; under severe infection conditions, glycerophospholipid metabolism and fatty acid degradation pathways are activated. These changes suggest that the products of photosynthesis may shift from meeting growth demands to synthesizing defensive substances (such as those in the phenylpropanoid metabolic pathway) and repairing membrane systems (phospholipid regeneration) [27]. Additionally, plant–pathogen interactions and plant hormone signaling pathways are significantly upregulated, indicating that after rust disease infection, *P. hendersonii* activates a series of complex defense responses to cope with pathogen invasion.

In the mild stage of infection, the PPO activity of *P. hendersonii* significantly increased by 134.4%, and the FRAP increased by 77.4% compared to the control, indicating that the phenolic oxidative defense system was efficiently activated. In the severe stage of infection, although PPO activity decreased to 159.1% of the control, the FRAP value further increased by 214.5%; the content of total phenols and flavonoids significantly decreased compared to the control group; and the SOD activity increased by 9% compared to the mild infection stage, indicating a significant improvement in antioxidant capacity in the later stage of rust infection. Under mild disease conditions, *P. hendersonii* may preferentially mediate non-cell-death-dependent defense by activating PPO, an enzyme that catalyzes the oxidation of phenolic substances such as chlorogenic acid to produce quinone toxins, directly inhibiting hyphal expansion and reducing substrate ROS production (O_2_^−^) [28,29]. The SOD activity was not significantly induced, which may be due to the weak ROS outbreak during the early stage of rust infection (the H_2_O_2_ content did not change significantly). During the severe infection stage, pathogens may trigger programmed cell death, leading to the inactivation of PPO located in the cytoplasm as the host cell disintegrates. At this stage, the activity of SOD is significantly increased and the total phenolic/flavonoid content is significantly reduced compared to the mild disease stage, which may be to clear excessive ROS to protect the tissue. In summary, *P. hendersonii* strengthens the cell wall by oxidizing phenolic substances through PPO, leading to early chemical defense; in the late stage of infection, plants rely on antioxidant enzymes such as SOD and non-enzymatic antioxidants such as total phenols/flavonoids to maintain oxidative stability and reduce tissue damage. This stage-specific defense strategy shift may be the result of balancing the allocation of disease-resistant resources in *P. hendersonii*.

Plant hormones play a crucial role in regulating defense mechanisms [30]. For example, SA activates the defense response of Populus against *Melampsora larici-populina* by enhancing the biosynthesis of catechins and proanthocyanidins [31]. In this study, during the mild infection stage, the SA content in *P. hendersonii* significantly increased to 11.9 times that of the control group, which often activates the plant’s systemic acquired resistance. Meanwhile, the abscisic acid (ABA) content decreased to 41.2% in the control group, potentially alleviating the inhibitory effect of ABA on defense-related genes [32]. However, during the severe infection stage, although the SA level decreased, it remained higher than that of the control group, while ABA levels returned to control levels. This shift may be attributed to the sustained high expression of SA leading to an excessive metabolic burden, indicating that the plant is transitioning from an active defense strategy to a tolerance strategy [33]. Additionally, the rise in ABA levels may limit the spread of pathogens by regulating stomatal closure and coordinating resource allocation [34,35]. Transcriptomic analysis revealed a significant upregulation of the linoleic acid synthesis pathway, but no accumulation of jasmonic acid (JA) and its derivatives was detected. We speculate that the peak of JA synthesis may have been missed at the sampling time, and high concentrations of SA antagonize the synthesis of JA [36].

The metabolism of phenylpropanoids is a core defense mechanism in plants responding to rust disease stress [37]. In this study, the activity of the key rate-limiting enzyme, PAL, continuously increased with the severity of susceptibility (with PAL activity increasing by 74.39% and 138.67% in mildly and severely infected leaves, respectively). This significantly drove the substantial accumulation of phenolic and flavonoid compounds during the early stages of infection, thereby enhancing the antibacterial and antioxidant capabilities of *P. hendersonii* [13]. In mildly infected leaves, *P. hendersonii* rapidly upregulates the expression of genes encoding caffeic acid COMT, POD, and CAD. COMT catalyzes the conversion of caffeic acid into ferulic acid (G/S lignin precursor), working in conjunction with CAD and POD to promote the synthesis of G-type lignin [38,39,40,41]. In the severe disease stage, *P. hendersonii* upregulates the expression of genes encoding F5H, an enzyme that directs the conversion of ferulic acid into 5-hydroxyferulic acid (a precursor of S-type lignin) [42]. This process facilitates the synthesis of S-type lignin and enhances the degree of cell wall lignification [43]. The variation in the types of genes encoding lignin synthesis-related enzymes at different levels of disease severity indicates that the types of lignin synthesized by *P. hendersonii* differ at various disease stages. The alteration in lignin type suggests that *P. hendersonii* strengthens its defense capabilities by increasing lignin strength. Additionally, the biosynthetic pathways of the cuticle and suberin layers are activated at different disease stages, indicating that the physical barrier is further reinforced, which is significant for preventing pathogen invasion.

Additionally, COMT can indirectly promote the biosynthesis of chlorogenic acid and neochlorogenic acid. As potent antioxidant defense molecules, chlorogenic acid [44] can inhibit disease progression by disrupting the cell membranes of pathogens [45,46]. This study found that the content of chlorogenic acid and neochlorogenic acid significantly increased during the mild infection stage, suggesting that COMT may drive the accumulation of these phenolic acid defense substances to combat rust disease infection. Notably, the levels of flavonoid glycosides such as hyperoside and rutin also significantly increased under mild infection conditions; however, they sharply declined under severe infection conditions. This may be attributed to the activation of starch and sucrose metabolism under mild infection conditions, which may promote glycosylation modifications and subsequently enhance their accumulation. In contrast, during the severe infection stage, the suppression of carbon metabolism pathways may lead to a decrease in sugar donors, thus limiting their accumulation [47]. In summary, *P. hendersonii* enhances physical barriers by upregulating genes related to lignin and cutin synthesis while increasing the accumulation of phenolic acids and flavonoid glycosides in response to rust disease invasion.

## 4. Materials and Methods

### 4.1. Experimental Materials

Samples of *Poacynum hendersonii* were collected from a standardized cultivation base located in Yuli County, Xinjiang, China (latitude 41.3°, longitude 86.2°; altitude 900 m; saline-alkali soil with a pH of 8.3). The reference standards were sourced as follows: chlorogenic acid (purity ≥ 98%), cryptochlorogenic acid (purity ≥ 98%), quercetin (purity ≥ 95%), and kaempferol (purity ≥ 97%) were purchased from Feng Biotechnology (Shanghai, China); astragaloside (purity ≥ 98%) and hyperoside (purity ≥ 98%) were obtained from Huilin Biotechnology (Xi’an, China); isorhamnetin (purity ≥ 95%) was procured from Bomei Biotechnology (Hefei, China); and rutin (purity ≥ 94%) was acquired from Solabio Technology (Beijing, China).

### 4.2. Experimental Methods

In July 2023, during the peak flowering period, three-year-old *P. hendersonii* were selected for this study (Figure 12A,B). The plants were divided into three groups: the control group (CK) which consisted of healthy plants without disease spots (Figure 12C) and two infected groups categorized based on the severity of symptoms according to the infection grading standard for *M. apocyni*: the mild disease group (M) and the severe disease group (S). In the mild disease group (M), lesions were located on the underside of the leaves, covering 5–10% of the total leaf area (Figure 12D); the severe disease group (S) exhibited lesions on both the upper and lower surfaces of the leaves, occupying 50–60% of the area, accompanied by significant yellowing (Figure 12E). Fifteen plants were randomly selected from each group, and three leaves were collected from each plant. These leaves were immediately wrapped in pre-cooled aluminum foil and quickly frozen in liquid nitrogen. The samples were then transferred to a −80 °C ultra-low temperature freezer for storage, to be used for physiological parameter measurement, transcriptome analysis, and quantitative PCR validation.

#### 4.2.1. Determination of Antioxidant Activity and Defense Enzyme Activity

Using kits provided by Beijing Solarbio Biotechnology Co., Ltd., the activities of superoxide dismutase (SOD), phenylalanine ammonia-lyase (PAL), and polyphenol oxidase (PPO) were measured in healthy and diseased leaves of *P. hendersonii*, along with the H_2_O_2_ content and total antioxidant capacity (FRAP). Each treatment was repeated three times, and the average value was calculated.

#### 4.2.2. Determination of Chlorophyll Content

Weigh 0.10 g of fresh leaves and chop them into small pieces of 1 mm × 1 mm [48]. Add 10 mL of the extraction solution, which consists of a 1:1 (*v*/*v*) mixture of acetone and anhydrous ethanol, and place the mixture in a water bath at 60 °C for thorough extraction until the tissue becomes completely white. After cooling to room temperature, adjust the volume to 10 mL. Centrifuge the mixture at 8000 rpm (SLX-1024F, Wuhan, Servicebio, China) for 10 min at 4 °C. Take 1.0 mL of the supernatant and measure the absorbance at 663 nm and 645 nm using a UV spectrophotometer (UV-1900, Shimadzu, China), denoted as A_663_ and A_645_, respectively. Calculate the contents of chlorophyll a and chlorophyll b using the following formulas, repeating the measurements five times to obtain average values: chlorophyll a (mg·g^−1^ FW) = [12.71 × A_663_ − 2.59 × A_645_] × V/(W × 1000) and chlorophyll b (mg·g^−1^ FW) = [22.88 × A_645_ − 4.67 × A_663_] × V/(W × 1000).

#### 4.2.3. Determination of Hormone Content

The hormone content was determined using an ultra-high performance liquid chromatography–triple quadrupole tandem mass spectrometry (UPLC-MS/MS, ACQUITY UPLC, Waters, USA) system. A total of 5.0 mg of salicylic acid (SA) and abscisic acid (ABA) standards (Solarbio, purity ≥ 98%) were weighed, dissolved in methanol, and diluted to a final volume of 5 mL to prepare a stock solution with a concentration of 1.0 mg·mL. This stock solution was further diluted to create standard working solutions at concentrations of 1, 10, 50, 100 and 200 ng·mL^−1^. Subsequently, 0.100 g of lyophilized leaf powder (ground in liquid nitrogen) was added to 5 mL of pre-cooled 80% methanol (containing 0.1% formic acid) and extracted in the dark at 4 °C for 10 h. The mixture was centrifuged at 12,000× *g* (4 °C) for 15 min to collect the supernatant; the precipitate was re-extracted with 200 μL of 80% methanol for an additional 10 h. After centrifugation, the supernatants were combined and filtered through a 0.22 µm PTFE filter membrane for future analysis. A standard curve was established with the concentration of the standard solution (x, ng·mL^−1^) plotted on the x-axis and the response peak area (y) on the y-axis: SA: y = 1782.18x − 402.849 (R^2^ = 0.998) and ABA: y = 3566.98x + 434.445 (R^2^ = 0.999). The hormone content in the samples (ng·g^−1^ DW) was calculated using this formula. Each treatment was repeated three times to compute the average value.

#### 4.2.4. Determination of Secondary Metabolites

The total flavonoid content in the leaves was determined using a spectrophotometric method [49], where 50.0 mg of rutin standard (Solebao Biotechnology Co., Ltd., Beijing, China) was dissolved in 50 mL of 60% ethanol solution, with solubilization assisted by a water bath at 40 °C. After cooling to room temperature, the solution was made up to volume. Aliquots of 0, 1, 2, 3, 4, and 5 mL of the rutin standard solution were transferred into volumetric flasks and diluted to 5 mL with distilled water. Subsequently, 0.3 mL of 5% NaNO_2_ solution was added, mixed, and allowed to stand for 6 min. Then, 0.3 mL of 10% Al(NO_3_)_3_ solution was added, mixed, and allowed to stand for another 6 min. Finally, 4.0 mL of 4% NaOH solution was added, and the volume was adjusted to 10 mL with distilled water. The mixture was then mixed thoroughly and allowed to stand in the dark for 20 min. The absorbance (x) of these solutions was measured at a wavelength of 510 nm using a UV–visible spectrophotometer. A regression equation for rutin concentration (y, mg·mL^−1^) as the dependent variable and absorbance as the independent variable was established: y = 8.0597 x + 0.0092 (R^2^ = 0.9997).Healthy and diseased leaves were dried at 60 °C until a constant weight was achieved, then thoroughly ground and passed through an 80-mesh sieve. Then, 0.400 g of leaf powder was transferred to a centrifuge tube, and 10 mL of 60% ethanol was added for ultrasonic extraction (KQ5200B, Kunshan Ultrasonic Instrument Co., Ltd., Kunshan, China) for 60 min. Subsequently, the mixture was centrifuged at 12,000 rpm for 10 min at 25 °C, and the supernatant was collected and diluted to a final volume of 10 mL for subsequent use. An aliquot of 1.0 mL of the supernatant was taken, and NaNO_2_, Al(NO_3_)_3_, and NaOH solutions were added sequentially, followed by dilution to a final volume of 10 mL in the dark for 20 min. The absorbance at 510 nm was measured, and the total flavonoid content was calculated based on the regression equation of rutin, expressed as rutin equivalent (mg·g^−1^). Each treatment was repeated three times, and the average value was calculated.The total phenolic content in the leaves was determined using the Folin–Ciocalteu method [50], where 50.0 mg of gallic acid standard (Soleibao Biotechnology Co., Ltd., Beijing, China) with a purity of ≥98% was weighed and dissolved in 50 mL of methanol to obtain a stock solution of 1 mg·mL^−1^. Subsequently, 0, 1, 2, 3, 4, and 5 mL of the stock solution were transferred into 10 mL volumetric flasks and diluted with methanol to obtain gallic acid standard solutions with concentrations of 0, 10, 20, 30, 40, and 50 mg·mL^−1^, respectively. The absorbance (x) of these solutions was measured at a wavelength of 760 nm using a UV-Vis spectrophotometer, and a regression equation was established with gallic acid concentration (y, mg·mL) as the dependent variable: y = 4.8332x − 0.045 (R^2^ = 0.9994).Weigh 0.10 g of powdered healthy leaves, mildly infected leaves, and severely infected leaves, and add 10 mL of a 5.5% methanol–hydrochloric acid solution. Then, perform ultrasonic extraction at 80 °C for 120 min, followed by centrifugation at 5,000 rpm for 20 min at 25 °C to collect the supernatant. Take 0.1 mL of the supernatant, add 5.0 mL of methanol and 0.1 mL of Folin–Ciocalteu reagent (Solaibao Biotechnology Co., Ltd., Beijing, China), mix thoroughly, and allow it to stand for 5 min. Next, add 0.2 mL of 20% Na_2_CO_3_ solution and allow it to stand in the dark for 30 min, followed by centrifugation at 12,000 rpm for 10 min. Take 1.0 mL of the supernatant and measure its absorbance at a wavelength of 760 nm. The total phenolic content is calculated using the regression equation of gallic acid, expressed as gallic acid equivalents (mg·g^−1^). Each sample solution is measured in triplicate, and the averages are calculated.Weigh 5.0 mg of standards including chlorogenic acid, cryptochlorogenic acid, quercetin, kaempferol, vitexin, hyperoside, isorhamnetin, and rutin (purity ≥ 98%) and dissolve them in methanol to a final volume of 5 mL, preparing a stock solution at a concentration of 1.0 mg·mL^−1^. The stock solution is then sequentially diluted to prepare standard solutions at concentrations of 1, 10, 100, 250, 500, 1000, 5000, and 10,000 ng·mL^−1^. The UPLC-MS/MS method is employed to measure the peak areas (y) of these eight compounds in the mixed solutions, establishing a regression equation with mass concentration (x, ng·mL) as the independent variable (Table 4). Weigh 0.500 g of powdered healthy and diseased leaves, add 5 mL of 60% (*v*/*v*) ethanol, and perform ultrasonic extraction for 120 min. The mixture is then centrifuged at 12,000 rpm for 20 min at room temperature. The supernatant is filtered through a 0.22 µm PTFE filter membrane, and 100 µL of the supernatant is transferred to a brown LC injection vial for UPLC-ESI-MS/MS analysis. The content of these eight substances is calculated based on the aforementioned regression equation, and the process is repeated three times to obtain the average value.Mass spectrometry conditions were established using an electrospray ionization source (ESI), with content determination conducted via multiple reaction monitoring mode (MRM). The desolvation gas temperature was maintained at 450 °C, while the ion source temperature was set to 150 °C. The desolvation gas flow rate was adjusted to 800 L·h^−1^, and the cone gas flow rate was specified at 150 L·h^−1^. The capillary voltage was configured to 3000 V (Table 4).The chromatographic conditions employed in this study include a mobile phase composed of phase A (0.1% formic acid aqueous solution) and phase B (acetonitrile). The analysis was conducted using a Waters ACQUITY UPLC BEH C18 column (50 mm × 2.1 mm, 1.7 μm) at a flow rate of 0.3 mL·min^−1^ and an injection volume of 1 μL. The column temperature was maintained at 40 °C, as detailed in Table 4.

#### 4.2.5. Transcriptome Sequencing and Analysis

Samples of healthy and diseased leaves (three biological replicates each) were rapidly frozen in liquid nitrogen, and total RNA was extracted using the Trizol method. The quality of RNA was verified through three tests: integrity was assessed by 1% agarose gel electrophoresis; purity was evaluated using a NanoPhotometer (OD260/280 = 1.8–2.2, OD260/230 ≥ 2.0); and the RIN value was confirmed to be ≥8.0 using an Agilent 2100 analyzer. Qualified samples (≥800 ng) were utilized to construct the library with the NEBNext^®^ Ultra™ RNA Library Prep Kit: mRNA was enriched using Oligo(dT) magnetic beads, fragmented using divalent cations (94 °C for 8 min), reverse transcribed to synthesize double-stranded cDNA, and underwent end repair, A-tailing, and ligation of Illumina adapters. The library was then purified using AMPure XP beads to select fragments of 200 ± 20 bp and amplified by PCR. After quantification with Qubit 4.0 and quality control with Agilent 2100 (insert size 350 ± 50 bp), sequencing was performed on the Illumina NovaSeq 6000 platform with a PE150 protocol. The original data underwent quality control using Fastp v0.23.2, which involved the removal of reads containing adapters, those with N bases exceeding 5%, and reads with Qphred scores ≤ 20 (with Q30 ≥ 90% and GC content between 40 and 60%). Clean reads were aligned to the *P. hendersonii* [51] reference genome using HISAT2 v2.0.5. Gene expression quantification was performed using featureCounts v2.0.3, normalized to FPKM. Differentially expressed genes (DEGs) were identified using DESeq2 v1.16.1 (|log_2_FC| ≥ 1.5 and *padj* < 0.05). Functional enrichment analysis was conducted based on Gene Ontology GO and Kyoto Encyclopedia of Genes and Genomes KEGG databases (*padj* < 0.05).

#### 4.2.6. Real-Time Quantitative Reverse Transcription PCR Analysis

This study commissioned Shanghai Lingen Company to verify the expression levels of five randomly selected differentially expressed genes (DEGs) relative to the reference gene Actin2 using reverse transcription quantitative polymerase chain reaction (RT-qPCR) technology. Each gene was analyzed with three biological replicates. The specific experimental procedure is as follows: First, the gDNA Eraser system was utilized to eliminate DNA contamination from total RNA samples. Second, the PrimeScript^®^ RT Enzyme Mix I was employed to the reverse transcription reaction, converting RNA into cDNA; subsequently, the qPCR reaction system was prepared using specific primers and qPCR Mix for amplification (Table 5). Finally, the qPCR reaction parameters were set, and data analysis was performed using the 2^−∆∆Ct^ method.

### 4.3. Data Analysis

Data processing was conducted using SPSS 25.0 (IBM Corp, Armonk, New York, NY, USA). All physiological and biochemical parameters are expressed as mean ± standard deviation (Mean ± SD). The differences between healthy leaves and those exhibiting varying degrees of disease (mild M/severe S) were analyzed using one-way ANOVA (*p* < 0.05), with significant results further assessed through Duncan’s multiple range test for inter-group comparisons. Data visualization was performed using OriginPro 2017 (OriginLab Corporation, Northampton, MA, USA), where the height of the bar charts represents the mean, and the error bars indicate the standard deviation.

## 5. Conclusions

This study reveals the response mechanism of *P. hendersonii* to rust disease by integrating physiological, biochemical, and transcriptomic data. With the increasing severity of the disease, rust significantly inhibits the photosynthesis of *P. hendersonii*, as evidenced by a sharp decline in chlorophyll a/b content and a notable suppression of carbon fixation. Under mild disease conditions, *P. hendersonii* regulates phenylpropane metabolism by activating the salicylic acid signaling pathway, promoting the accumulation of phenolic and flavonoid substances, accelerating lignin deposition, and possibly relying on PPO-mediated quinone product synthesis to synergistically enhance defense capabilities. Under severe disease conditions, *P. hendersonii* increases SOD activity and consumes phenolic and flavonoid compounds to enhance antioxidant capacity, while turning to the synthesis of highly resistant S-type lignin to build a stronger physical barrier. This study clarifies the response process of *P. hendersonii* to rust disease and reveals its resistance mechanisms, with key enzyme genes (such as COMT, POD, CAD, and F5H) identified as crucial targets for disease-resistant breeding.

## Figures and Tables

**Figure 1 plants-14-02589-f001:**
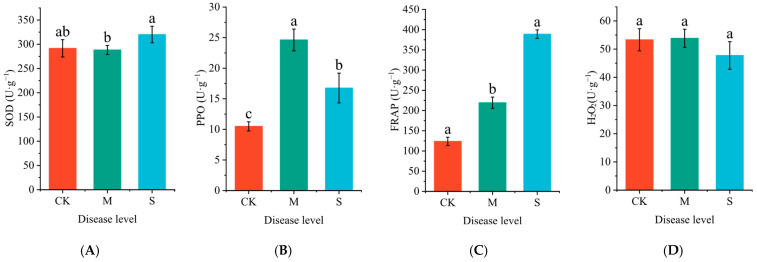
Effects of rust disease on the activities of superoxide dismutase (SOD, (**A**)), polyphenol oxidase (PPO, (**B**)), total antioxidant capacity (FRAP, (**C**)), and hydrogen peroxide (H_2_O_2_, (**D**)) content in the leaves of *P. hendersonii*. Note: Different lowercase letters represent significant differences between treatments at the *p* < 0.05 level.

**Figure 2 plants-14-02589-f002:**
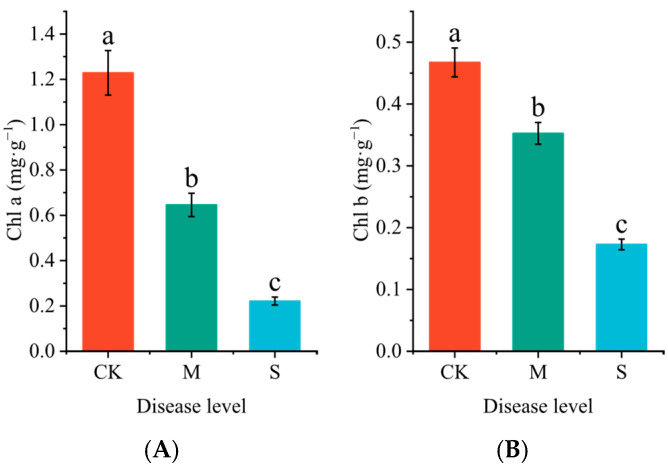
Effects of Rust on chlorophyll a (Cha, (**A**)) and chlorophyll b (Chb, (**B**)) content in the leaves of *P. hendersonii*. Different lowercase letters represent significant differences between treatments at the *p* < 0.05 level.

**Figure 3 plants-14-02589-f003:**
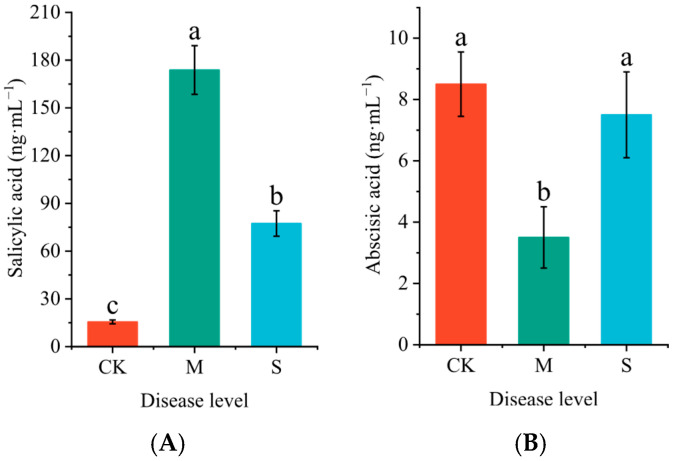
Effect of rust disease on salicylic acid (SA, (**A**)) and abscisic acid (ABA, (**B**)) content in leaves of *P. hendersonii*. Different lowercase letters represent significant differences between treatments at the *p* < 0.05 level (as shown in the same figure below).

**Figure 4 plants-14-02589-f004:**
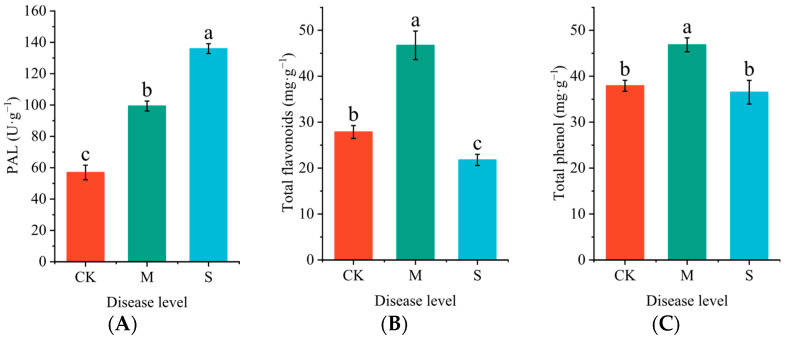
Effects of rust disease on activity of L-phenylalanine ammonia lyase (PAL, (**A**)), content of total flavonoids (**B**), and total phenols (**C**) in leaves of *P. hendersonii*. Different lowercase letters represent significant differences between treatments at the *p* < 0.05 level (as shown in the same figure below).

**Figure 5 plants-14-02589-f005:**
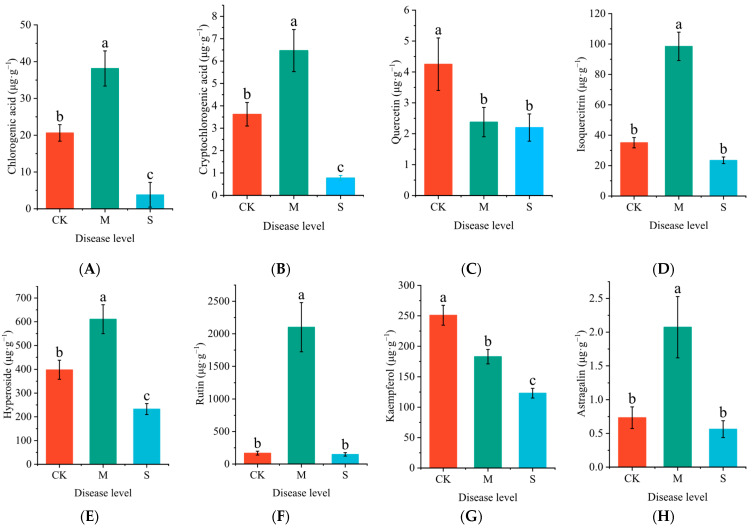
Effect of rust disease on the characteristic secondary metabolites in leaves of *P. hendersonii*. Chlorogenic acid (**A**), cryptochlorogenic acid (**B**), quercetin (**C**), isoquercetin (**D**), hypericin (**E**), rutin (**F**), kaempferol (**G**) and astragaloside (**H**). Different lowercase letters represent significant differences between treatments at the *p* < 0.05 level (as shown in the same figure below).

**Figure 6 plants-14-02589-f006:**
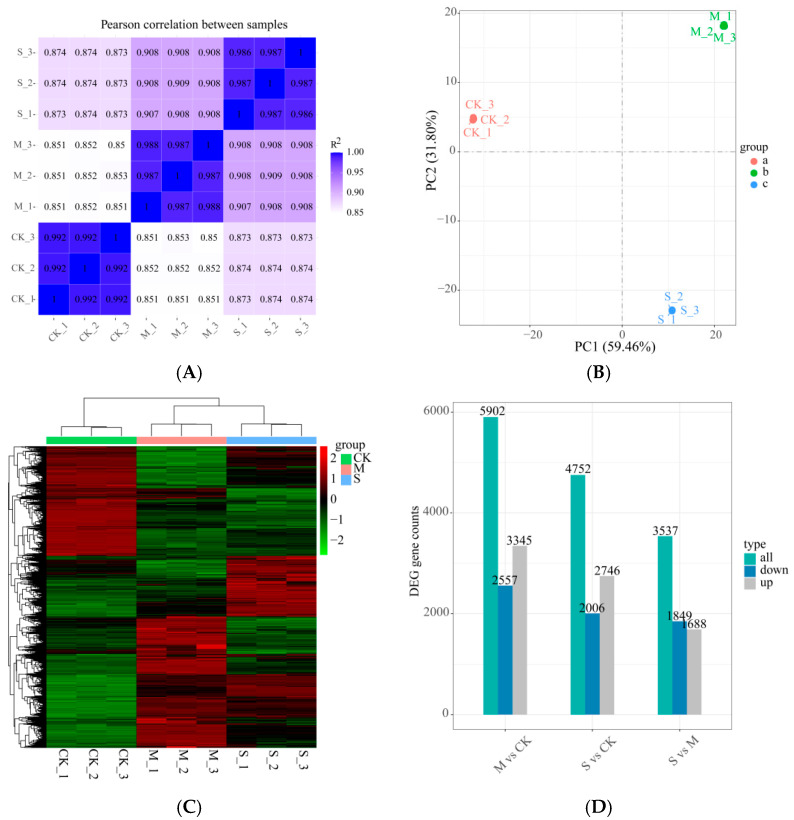
(**A**): Heatmap of Pearson correlation coefficients (the closer the Pearson correlation coefficient R^2^ is to 1, the smaller the differences between repeated samples of the same treatment); (**B**): Principal Component Analysis (PCA) plot (the horizontal axis PC1 and vertical axis PC2 represent the scores of the first and second principal components, respectively, with different colored scatter points indicating samples from different experimental groups); (**C**): clustering heatmap, where the vertical axis represents the clustering of samples, with shorter branches indicating higher similarity; (**D**): bar chart of DEGs.

**Figure 7 plants-14-02589-f007:**
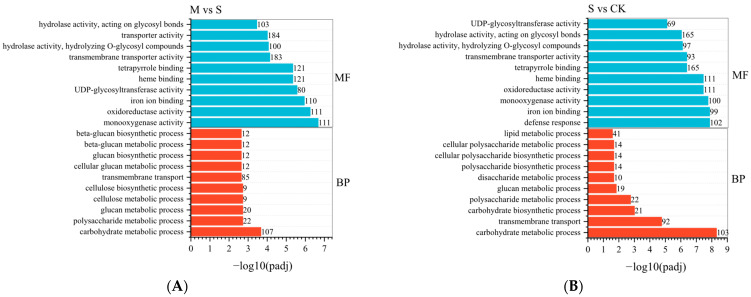
GO analysis of differentially expressed genes under mild (**A**) and severe (**B**) disease conditions. *y*-axis: GO terms grouped by ontology categories (biological process, molecular function, cellular component); *x*-axis: −log_10_(*padj*) enrichment score; Number: number of differentially expressed genes.

**Figure 8 plants-14-02589-f008:**
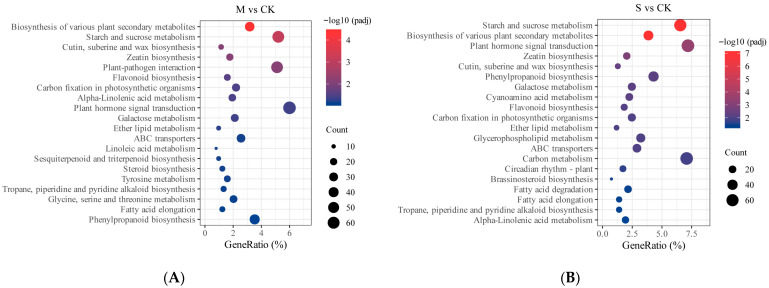
KEGG enrichment analysis of differentially expressed genes under mild (**A**) and severe (**B**) disease conditions. Note: The horizontal axis in the figure represents x/y (the number of differential genes in the corresponding metabolic pathway/the total number of genes identified in that pathway); the color of the dots indicates the −log_10_(*padj*); the size of the dots represents the number of differential metabolites in the corresponding pathway.

**Figure 9 plants-14-02589-f009:**
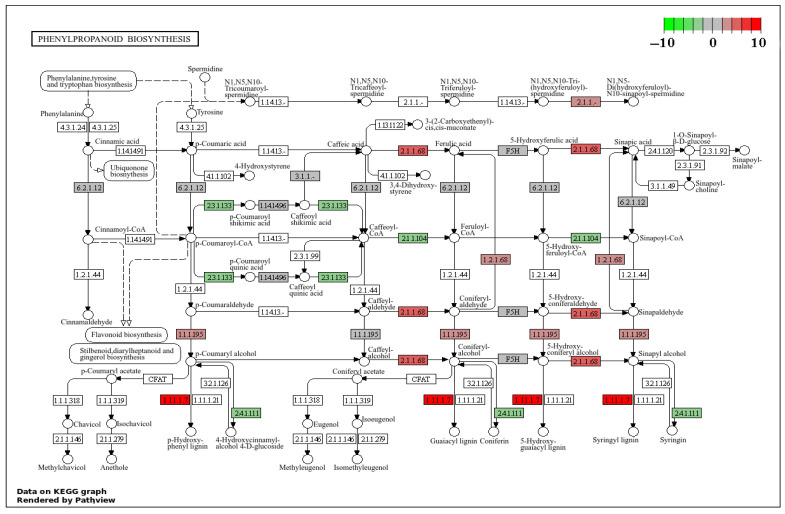
Biosynthetic pathway of phenylpropanoids under mild disease conditions. Note: The color represents the log_2_(FC) of gene expression, with red to green indicating a decrease in expression levels from high to low.

**Figure 10 plants-14-02589-f010:**
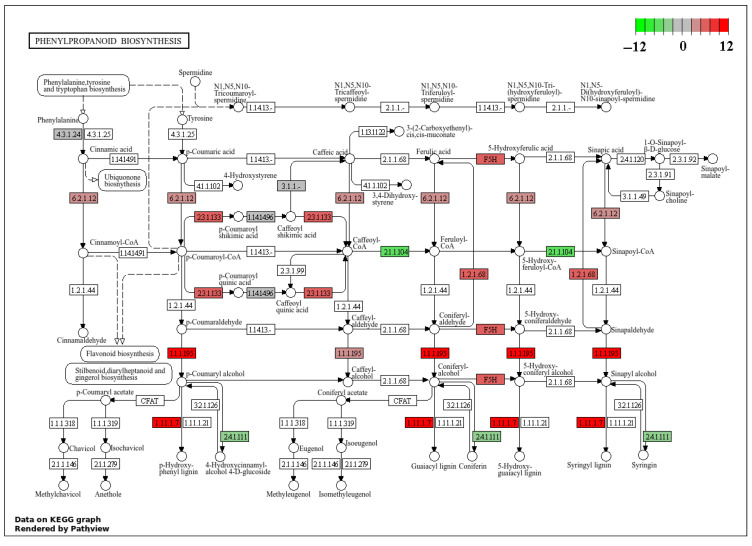
Biochemical pathway of phenylalanine biosynthesis under severe disease conditions. Note: The colors represent the fold change of gene expression log_2_(FC), with red to green indicating expression levels from high to low.

**Figure 11 plants-14-02589-f011:**
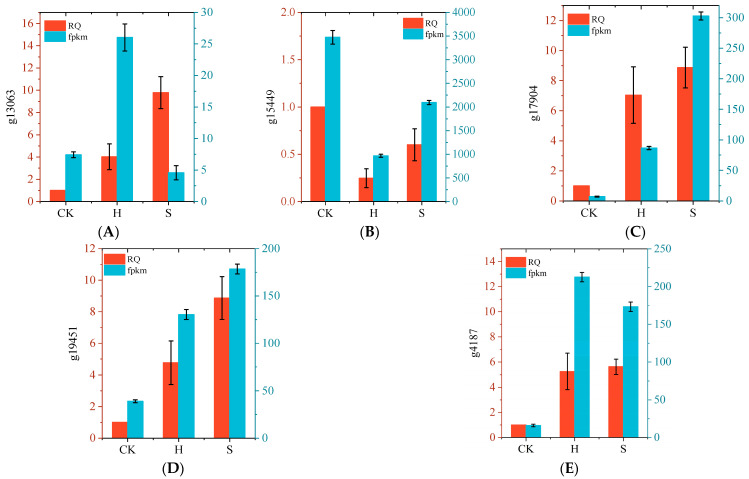
Verification of RNA seq results by qRT PCR. g13063 (**A**), g15449 (**B**), g17904 (**C**), g19451 (**D**), g4187 (**E**).

**Figure 12 plants-14-02589-f012:**
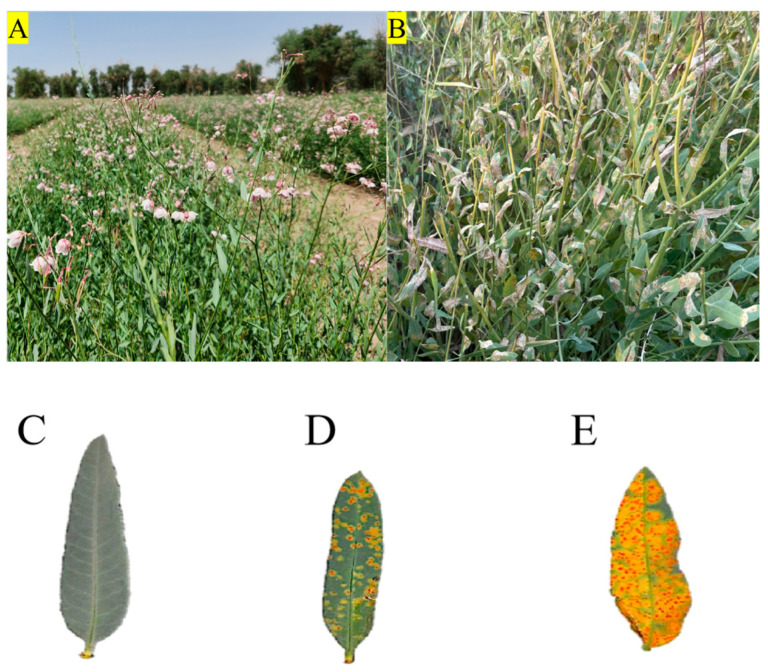
(**A**): Cultivated *P. hendersonii*; (**B**): plants infected with rust disease; (**C**): healthy leaves; (**D**): leaves with mild infection; (**E**): leaves with severe infection.

**Table 1 plants-14-02589-t001:** Transcriptome quality assessment.

Sample	Raw (Reads)	Raw_Bases (G)	Clean (Reads)	Clean_Bases (G)	Error_Rate (%)	Q20 (%)	Q30 (%)	GC_pct (%)
CK_1	21,213,008	6.4	20,764,146	6.2	0.03	97.75	93.57	43.85
CK_2	22,013,756	6.6	21,448,738	6.4	0.03	97.67	93.43	43.84
CK_3	21,793,993	6.5	21,114,531	6.3	0.03	97.67	93.39	43.87
M_1	20,722,380	6.2	20,285,020	6.1	0.02	98.1	94.45	44.32
M_2	22,825,204	6.8	22,153,576	6.6	0.02	98.04	94.29	44.44
M_3	20,829,249	6.2	20,386,203	6.1	0.03	97.94	94.04	44.32
S_1	19,765,517	5.9	19,295,848	5.8	0.02	98.15	94.53	45.06
S_2	20,893,270	6.3	20,258,248	6.1	0.02	98	94.18	45.15
S_3	19,844,042	6	19,259,470	5.8	0.02	98.15	94.55	45.11

**Table 2 plants-14-02589-t002:** Enriched KEGG pathways for differentially expressed genes in mild disease status.

KEGGID	Description	*p* Value	*padj*	Up	Down
ath00999	Biosynthesis of various plant secondary metabolites	0.000	0.000	21	15
ath00500	Starch and sucrose metabolism	0.000	0.001	34	25
ath00073	Cutin, suberine and wax biosynthesis	0.000	0.008	9	4
ath00908	Zeatin biosynthesis	0.000	0.008	10	10
ath04626	Plant-pathogen interaction	0.000	0.008	42	16
ath00941	Flavonoid biosynthesis	0.001	0.026	3	15
ath00710	Carbon fixation in photosynthetic organisms	0.002	0.034	4	21
ath00592	alpha-Linolenic acid metabolism	0.002	0.034	14	8
ath04075	Plant hormone signal transduction	0.003	0.043	38	30
ath00052	Galactose metabolism	0.003	0.043	18	6
ath00565	Ether lipid metabolism	0.005	0.053	9	2
ath02010	ABC transporters	0.007	0.066	17	12
ath00591	Linoleic acid metabolism	0.008	0.066	8	1
ath00909	Sesquiterpenoid and triterpenoid biosynthesis	0.008	0.066	4	7
ath00100	Steroid biosynthesis	0.008	0.066	3	11
ath00350	Tyrosine metabolism	0.008	0.066	8	10
ath00960	Tropane, piperidine and pyridine alkaloid biosynthesis	0.009	0.069	5	10
ath00260	Glycine, serine and threonine metabolism	0.010	0.071	8	15
ath00062	Fatty acid elongation	0.012	0.080	4	10
ath00940	Phenylpropanoid biosynthesis	0.015	0.096	23	17

**Table 3 plants-14-02589-t003:** Enrichment of differentially expressed genes in KEGG pathways under severe disease conditions.

KEGGID	Description	*p* Value	*padj*	Up	Down
ath00500	Starch and sucrose metabolism	0.000	0.000	28	33
ath00999	Biosynthesis of various plant secondary metabolites	0.000	0.000	22	14
ath04075	Plant hormone signal transduction	0.000	0.000	35	32
ath00908	Zeatin biosynthesis	0.000	0.001	9	10
ath00073	Cutin, suberine and wax biosynthesis	0.000	0.004	9	3
ath00940	Phenylpropanoid biosynthesis	0.000	0.006	25	15
ath00052	Galactose metabolism	0.000	0.006	15	8
ath00460	Cyanoamino acid metabolism	0.000	0.006	11	10
ath00941	Flavonoid biosynthesis	0.001	0.008	4	13
ath00710	Carbon fixation in photosynthetic organisms	0.001	0.009	7	16
ath00565	Ether lipid metabolism	0.001	0.009	9	2
ath00564	Glycerophospholipid metabolism	0.001	0.009	21	9
ath02010	ABC transporters	0.001	0.009	17	10
ath01200	Carbon metabolism	0.002	0.013	23	43
ath04712	Circadian rhythm—plant	0.003	0.022	3	13
ath00905	Brassinosteroid biosynthesis	0.004	0.031	7	0
ath00071	Fatty acid degradation	0.005	0.037	12	8
ath00062	Fatty acid elongation	0.006	0.042	6	7
ath00960	Tropane, piperidine and pyridine alkaloid biosynthesis	0.009	0.056	6	7
ath00592	alpha-Linolenic acid metabolism	0.010	0.062	11	7

**Table 4 plants-14-02589-t004:** Optimized mass spectrometry conditions, regression equations and correlation coefficients for 8 compounds.

Secondary Metabolite	Parent Ion (m·z^−1^)	Daughter Ion (m·z^−1^)	Ionization Mode	Voltage (V)	Collisional Energy (eV)	Retention Time (min)	Regression Equation	R^2^	Linear over (ng·mL^−1^)
Chlorogenic Acid	352.9	190.9 *	−	40	22	2.57	Y = 194.3 * X − 11.5	0.9996	0.9–5043.2
		84.9		40	36				
Cryptochlorogenic Acid	256.9	173.0 *	+	40	25	2.75	Y = 79.0 * X − 176.3	0.9992	11.9–5056.3
		190.9		40	22				
Quercetin	301.1	150.9 *	−	4	24	7.29	Y = 353.3 * X + 8370.9	0.9991	1.3–5035.6
		178.9		4	18				
Kaempferol	287.0	152.9 *	+	48	32	7.43	Y = 1620.1 * X + 50674.8	0.9994	1.1–4983.1
		121.0		48	30				
Astragalin	446.9	254.9	−	18	38	6.87	Y = 501.2 * X + 666.2	0.9999	10.1–5000.3
		284.0 *		18	24				
Hyperoside	463.1	300.2 *	−	8	26	5.98	Y = 253.9 * X − 1964.7	0.9990	12.5–5069.6
		270.9		8	40				
Isoquercitrin	463.0	300.2 *	−	66	28	6.30	Y = 277.1 * X − 977.4	0.9992	11.5–5060.1
		270.9		66	40				
Rutin	609.2	300.2 *	−	8	36	5.94	Y = 111.2 * X − 183.2	0.9993	11.6–5067.5
		255.0		8	52				

Note: “*” represents quantitative ion, R^2^ represents correlation coefficient.

**Table 5 plants-14-02589-t005:** Primers used for qPCR reaction.

Amplification Region	Primer Name	Primer Sequence
g15449	g15449-F	GTGATATGTGCTCTAAGGATCTGG
	g15449-R	CTATGCCACTTCCAGCTATTATTT
g19451	g19451-F	GTTTCTTCAAAATCTATGCTGCTT
	g19451-R	TTCTTCCCAAAGGAATTTCATAAT
g4187	g4187-F	GTTAGGGTAGGGGATAAAGTAGGT
	g4187-R	CACCAAGTTTTACTAAGGCTTCCT
g13063	g13063-F	TGGGTGTTGTGGTGGAGTTA
	g13063-R	TTCACCTTGCTACAGTCGGT
g17904	g17904-F	AAGTCCGCAGTAGAGAGAGTGTGT
	g17904-R	TTTCTTGTTTGAGCTAAAGCAGTG
Actin2	Actin2-F	TGCTGGATTCTGGTGATGGT
	Actin2-R	AATTTCCCGCTCTGCTGTTG

## Data Availability

The sequencing data have been submitted to the SRA database under the accession number SUB15470142.

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
