# Peer review of "Physiological, Biochemical, and Molecular Mechanisms of Resistance of Poacynum hendersonii to Melampsora apocyni"

_plants, 2025, doi:10.3390/plants14162589_

Round 1

Reviewer 1 Report

Comments and Suggestions for Authors

Here are a few comments that may contribute to improving the manuscript:

Line 52: “apocyni > italics

Lines 122, 165: In the case of centrifugation, RPM alone is not sufficient. Either the rotor radius must also be specified, or the centrifuge model must be provided

Line 141: “organic filter membrane” > It is not sufficient to indicate the type of membrane; it's too generic. More detailed information is needed.

Lines 149, 174: “a precise amount” > Why specify 'precise'? In a scientific experiment, the weight should be exactly as stated.

Lines 150, 153: Why specify 'volumetric flask'? It's sufficient to indicate the “final volume” to which the liquid is adjusted.

Lines 175, 183: “accurately weighed”  > see note for lines 149 and 174.

Various lines: ultrasonic extractions > Specify conditions and instrumentation

Line 197: mg.ml    > The unit should be expressed as mL⁻¹; the -1 exponent is missing

Table 1: The paired rows for each individual compound are not clearly separated, which may make the data harder to read or less intuitive.

Lines 265, 280, 288, 299: If the paragraph title style requires italics, the Latin names of species should be in regular (non-italic) font

Table 3: The units of measurement are missing in the column headers

Line 334: “Macadamia integrifolia > italics

Table captions: Check that the style used to refer to the table is always consistent (e.g., the period after the number)

Discussion: Some of the opening sentences about the effects of hyphal growth on plant cells are obvious and can be removed (e.g. The research indicates that the hyphal invasion of the rust fungus disrupts the structure of mesophyll cells). The discussion of the results could be expanded by including comparisons with other host-pathogen pairs, commenting on whether the influence on the expression of various genes is similar or not. Otherwise, the focus remains limited to this single disease, which is restrictive. Common aspects with other pathogen-plant interactions or peculiarities of the one studied should be highlighted

Author Response

Dear reviewers and editors.

Thank you for giving us the opportunity to revise our manuscript titled “Physiological, biochemical and molecular mechanisms of resistance to rust in Poacynum hendersonii” (plants-3803735). We sincerely appreciate the constructive comments from the reviewers and the editorial team, which have significantly improved the quality of this work.

We have carefully revised the manuscript to enhance its quality and facilitate the understanding of the readers. Our point-to-point responses are presented in the following. We hope that the revision would satisfactorily address the comments and·concerns of the editors and reviewers.

Responses to Reviewers’ Comments

Comment 1: Line 52: “apocyni > italics.

Response 1: We agree with this comment and have made modifications.

Comment 2: Lines 122, 165: In the case of centrifugation, RPM alone is not sufficient. Either the rotor radius must also be specified, or the centrifuge model must be provided. 

Response 2: We sincerely thank the reviewer for pointing out the need to improve the centrifuge model. We have carefully revised the text and added the model of the centrifuge as follows:

1.Page 4, Line 122

Original: Centrifuge the mixture at 8000 rpm for 10 min at 4°C.

Revised: Centrifuge the mixture at 8000 rpm (SLX-1024F, Wuhan, Servicebio, China) for 10 min at 4°C.

Comment 3: Line 141: “organic filter membrane” It is not sufficient to indicate the type of membrane; it's too generic. More detailed information is needed.

Response 3: We sincerely thank the reviewer for pointing out the need to supplement the organic filter membrane model. We added the model of the organic filter membrane at the location where it first appeared in the manuscript

Original: through a 0.22 μm organic filter membrane for future analysis.

Revised: through a 0.22 µm PTFE filter membrane for future analysis.

Comment 4: Lines 149, 174, 175, 183: “a precise amount” > Why specify 'precise'? In a scientific experiment, the weight should be exactly as stated.

Response 4: We sincerely thank the reviewer for pointing out the grammatical errors. We have carefully revised the text as follows:

  1. Page 4,Lines 149

Original: A precise amount of 50.0 mg of rutin standard (Solebao Biotechnology Co., Ltd., China) was dissolved in 60% ethanol in a 50 mL volumetric flask, with solubilization assisted by a water bath at 40°C.

Revised: 50.0 mg of rutin standard (Solebao Biotechnology Co., Ltd., China) was dissolved in 50 mL of 60% ethanol solution, with solubilization assisted by a water bath at 40°C.

  1. Page 5,Lines 174, 175

Original: A precise amount of 50.0 mg of gallic acid standard (Soleibao Biotechnology Co., Ltd., China) with a purity of ≥ 98% was accurately weighed and transferred to a 50 mL volumetric flask, where it was prepared as a stock solution with a mass concentration of 1 mg·mL.

Revised: 50.0 mg of gallic acid standard (Soleibao Biotechnology Co., Ltd., China) with a purity of ≥ 98% was weighed and dissolved in 50 mL of methanol to obtain a stock solution of 1 mg·mL-1.

  1. Page 5,Lines 183

Original: Accurately weigh 0.10 g of powdered healthy leaves, mildly infected leaves, and se-verely infected leaves, and add 10 mL of a 5.5% methanol-hydrochloric acid solution.

Revised: Weigh 0.10 g of powdered healthy leaves, mildly infected leaves, and severely infected leaves, and add 10 mL of a 5.5% methanol-hydrochloric acid solution.

Comment 5: Lines 150, 153: Why specify 'volumetric flask'? It's sufficient to indicate the “final volume” to which the liquid is adjusted.

Response 5: We sincerely thank the reviewer for pointing out the methodological errors. We have carefully revised the manuscript as follows:

  1. Page 4,Lines 150

Original: A precise amount of 50.0 mg of rutin standard (Solebao Biotechnology Co., Ltd., China) was dissolved in 60% ethanol in a 50 mL volumetric flask, with solubilization assisted by a water bath at 40°C.

Revised: 50.0 mg of rutin standard (Solebao Biotechnology Co., Ltd., China) was dissolved in 50 mL of 60% ethanol solution, with solubilization assisted by a water bath at 40°C.

  1. Page 4,Lines 153

Original: Aliquots of 0, 1, 2, 3, 4, and 5 mL of the rutin standard solution were transferred into 10 mL volumetric flasks, and distilled water was added to achieve a final volume of 5 mL.

Revised: Aliquots of 0, 1, 2, 3, 4, and 5 mL of the rutin standard solution were transferred into volumetric flasks, and dilute to 5 mL with distilled water.

Comment 6: Various lines: ultrasonic extractions > Specify conditions and instrumentation.

Response 6: We sincerely thank the reviewer for pointing out the lack of ultrasonic extraction conditions and models in the manuscript. We added the instrument model at the location where the ultrasound extraction step first appeared in the manuscript.

Original: 0.400 g of leaf powder was transferred to a centrifuge tube, and 10 mL of 60% ethanol was added for ultrasonic extraction for 60 min.

Revised: 0.400 g of leaf powder was transferred to a centrifuge tube, and 10 mL of 60% ethanol was added for ultrasonic extraction (KQ5200B, Kunshan Ultrasonic Instrument Co., Ltd, China) for 60 min.

Comment 7: The article format needs to be revised.

Response 7: We sincerely thank the reviewers for their suggestions on the manuscript format. As recommended, we have revised the article to:

The formatting issues of lines 197, 265, 280, 288, 299, and 334, as well as Table 1 and the Table 3 and Table captions, have been modified.

Comment 8: Discussion: Some of the opening sentences about the effects of hyphal growth on plant cells are obvious and can be removed (e.g. The research indicates that the hyphal invasion of the rust fungus disrupts the structure of mesophyll cells). The discussion of the results could be expanded by including comparisons with other host-pathogen pairs, commenting on whether the influence on the expression of various genes is similar or not. Otherwise, the focus remains limited to this single disease, which is restrictive. Common aspects with other pathogen-plant interactions or peculiarities of the one studied should be highlighted.

Response 8:We appreciate this suggestion. The mentioned sentences about hyphal impacts on mesophyll cells have been removed. As our study focuses specifically on P. hendersonii's rust-resistance mechanisms (defense-related genes/metabolites), we did not prioritize photosynthetic analyses. 

We thank the reviewer for this insightful feedback again.

These additions pinpoint knowledge gaps while anchoring the research rationale to empirical precedents and unresolved challenges, significantly elevating the Introduction’s precision. We deeply appreciate the reviewer’s guidance in strengthening this section.

We believe these revisions have thoroughly addressed all concerns raised by the reviewers. Please do not hesitate to contact us if further clarifications are needed.

Thank you for your time and consideration.

Reviewer 2 Report

Comments and Suggestions for Authors

I am thankful for the opportunity to review this work. The manuscript has been thoroughly reviewed and meets the standards for publication. The research is original, well-structured, and makes a meaningful contribution to the field. I recommend acceptance for publication with some improvements.

1. The title is somewhat long and dense. Please reorganize without altering the meaning. Additionally, the causal agent of rust (Melampsora apocyni) is listed in the keywords but not mentioned in the title. Please include it in the title.

2. In the abstract, write the full scientific name of the pathogen at first citation. Moreover, the final sentence could be strengthened by adding a clear concluding statement.

3. In the ‘Introduction’ section, some information is repeated and should be condensed for clarity. The statement that “previous studies have only focused on pathogen identification” lacks specificity. Please cite at least one such study or briefly outlining the typical limitations of these earlier works. Additionally, while the final paragraph lists all the parameters being studied, it does not present a concise central hypothesis or clearly stated research aim. Please include a focused research objective would strengthen the introduction.

4. Section 3.1, ‘The Effect of Rust Disease on the Antioxidant Activity in Leaves of P. hendersonii,’ could benefit from deeper mechanistic insight. For example, why does PPO activity increase more than SOD during mild infection? What might account for the decline in PPO activity under severe infection? In addition, please include a brief statement linking these antioxidant responses to disease resistance or tolerance would help contextualize the importance of these findings within plant pathology.

5. In the ‘3.2. The effect of rust disease on chlorophyll content in leaves of P. hendersonii’ section, it is stated in the methods that ANOVA and Duncan’s test were used; however, this section does not explicitly mention statistical significance, such as p-values or letter groupings in the figure. Please clarify which differences are statistically significant. Although the results are clear, the section would be more impactful if it briefly cross-referenced transcriptomic findings related to photosynthesis—such as the downregulation of genes involved in carbon fixation pathways, as noted in the discussion. Additionally, including the chlorophyll a/b ratio or total chlorophyll content would provide further insights into how rust infection affects light-harvesting efficiency and overall photosynthetic performance.

6. In the ‘3.3. The effect of rust disease on hormone content in leaves of P. hendersonii’ section, only salicylic acid (SA) and abscisic acid (ABA) are analyzed. It would be beneficial to include or at least mention other stress-related hormones such as jasmonic acid (JA), ethylene (ET), or auxins, particularly since the transcriptomic analysis broadly references hormone signaling pathways.

7. In ‘Discussion’ section, several statements e.g., prioritizing PPO over SOD, transition to tolerance strategy are plausible but lack references. Please add references for defense strategy transitions and enzymatic preferences. Additionally, the absence of jasmonic acid is mentioned, however, alternative explanations e.g., detection limits or post-translational regulation are not considered. Please discuss other potential factors or technical limitations in not detecting JA.

Author Response

Dear reviewers and editors.

Thank you for giving us the opportunity to revise our manuscript titled “Physiological, biochemical and molecular mechanisms of resistance to rust in Poacynum hendersonii” (plants-3803735). We sincerely appreciate the constructive comments from the reviewers and the editorial team, which have significantly improved the quality of this work.

We have carefully revised the manuscript to enhance its quality and facilitate the understanding of the readers. Our point-to-point responses are presented in the following. We hope that the revision would satisfactorily address the comments and·concerns of the editors and reviewers.

Responses to Reviewers’ Comments

Comment 1: The title is somewhat long and dense. Please reorganize without altering the meaning. Additionally, the causal agent of rust (Melampsora apocyni) is listed in the keywords but not mentioned in the title. Please include it in the title.

Response 1: We agree with this comment and have made changes to the title based on your suggestions without altering the original intention.

Original: Physiological, biochemical and molecular mechanisms of resistance to rust in Poacynum hendersonii

Revised: Physiological, biochemical, and molecular mechanisms of resistance of Poacynum hendersonii to Melampsora apocyni

Comment 2: In the abstract, write the full scientific name of the pathogen at first citation. Moreover, the final sentence could be strengthened by adding a clear concluding statement.

Response 2: We agree with your comment. We added the full name of the species when first citing it in the abstract and made a modification to the last sentence.

The rust disease caused by Melampsora apocyni seriously affects the growth of Poacynum hendersonii.

Our research provides important reference for the prevention and control of M. apocyni in P. hendersonii.

Comment 3: In the‘Introduction’section, some information is repeated and should be condensed for clarity. The statement that “previous studies have only focused on pathogen identification” lacks specificity. Please cite at least one such study or briefly outlining the typical limitations of these earlier works. Additionally, while the final paragraph lists all the parameters being studied, it does not present a concise central hypothesis or clearly stated research aim. Please include a focused research objective would strengthen the introduction.

Response 3: We agree with your comment and have appropriately condensed the content of the introduction to make the logic clearer. According to your suggestion, references have been added in the corresponding locations. Based on your suggestion, rewrite the last paragraph of the introduction to clearly state the research objectives of this study.

Comment 4: Section 3.1, ‘The Effect of Rust Disease on the Antioxidant Activity in Leaves of P. hendersonii,’ could benefit from deeper mechanistic insight. For example, why does PPO activity increase more than SOD during mild infection? What might account for the decline in PPO activity under severe infection? In addition, please include a brief statement linking these antioxidant responses to disease resistance or tolerance would help contextualize the importance of these findings within plant pathology.

Response 4: We agree with your comment. We further explained in the second paragraph of the discussion section why PPO increases more than SOD under mild disease conditions and decreases under severe disease conditions, and linked these changes to plant disease resistance.

Comment 5: In the ‘3.2. The effect of rust disease on chlorophyll content in leaves of P. hendersonii’ section, it is stated in the methods that ANOVA and Duncan’s test were used; however, this section does not explicitly mention statistical significance, such as p-values or letter groupings in the figure. Please clarify which differences are statistically significant. Although the results are clear, the section would be more impactful if it briefly cross-referenced transcriptomic findings related to photosynthesis—such as the downregulation of genes involved in carbon fixation pathways, as noted in the discussion. Additionally, including the chlorophyll a/b ratio or total chlorophyll content would provide further insights into how rust infection affects light-harvesting efficiency and overall photosynthetic performance.

Response 5: We agree with your comment and have highlighted the statistical significance results in section 3.2. Since we only presented the experimental results in the Results section of the paper, we combined the results of chlorophyll with the downregulation of the transcriptome "photosynthetic carbon sequestration pathway" in the Discussion section to enhance persuasiveness. Furthermore, we believe that the impact of rust on the chlorophyll content of large leaved white hemp is evident, therefore chlorophyll a and chlorophyll b are sufficient to support the conclusion.

Comment 6:  In the ‘3.3. The effect of rust disease on hormone content in leaves of P. hendersonii’ section, only salicylic acid (SA) and abscisic acid (ABA) are analyzed. It would be beneficial to include or at least mention other stress-related hormones such as jasmonic acid (JA), ethylene (ET), or auxins, particularly since the transcriptomic analysis broadly references hormone signaling pathways.

Response 6:We thank the reviewer for highlighting this important aspect. While our original study focused on SA and ABA due to their established roles in biotrophic pathogen resistance, we fully agree that JA/ET/IAA are critical for stress responses.

Comment 7: In ‘Discussion’ section, several statements e.g., prioritizing PPO over SOD, transition to tolerance strategy are plausible but lack references. Please add references for defense strategy transitions and enzymatic preferences. Additionally, the absence of jasmonic acid is mentioned, however, alternative explanations e.g., detection limits or post-translational regulation are not considered. Please discuss other potential factors or technical limitations in not detecting JA.

Response 7:We sincerely appreciate the reviewer's constructive suggestions. We have revised the Discussion section. We have added references in the corresponding locations and provided other explanations for the absence of jasmonic acid.

We thank the reviewer for this insightful feedback again.

These additions pinpoint knowledge gaps while anchoring the research rationale to empirical precedents and unresolved challenges, significantly elevating the Introduction’s precision. We deeply appreciate the reviewer’s guidance in strengthening this section.

We believe these revisions have thoroughly addressed all concerns raised by the reviewers. Please do not hesitate to contact us if further clarifications are needed.

Thank you for your time and consideration.

Round 2

Reviewer 2 Report

Comments and Suggestions for Authors

The authors have adequately addressed my comments.